# MALDI-TOF Mass Spectrometry Online Identification of *Trichophyton indotineae* Using the MSI-2 Application

**DOI:** 10.3390/jof8101103

**Published:** 2022-10-19

**Authors:** Anne-Cécile Normand, Alicia Moreno-Sabater, Arnaud Jabet, Samia Hamane, Geneviève Cremer, Françoise Foulet, Marion Blaize, Sarah Dellière, Christine Bonnal, Sébastien Imbert, Sophie Brun, Ann Packeu, Stéphane Bretagne, Renaud Piarroux

**Affiliations:** 1Service de Parasitologie-Mycologie, Hôpital La Pitié-Salpêtrière, AP-HP, 75013 Paris, France; 2Service de Parasitologie-Mycologie, Hôpital Saint-Antoine, AP-HP, 75012 Paris, France; 3Centre d’Immunologie et des Maladies Infectieuses, (CIMI-PARIS), Inserm U1135, Sorbonne Université, 75013 Paris, France; 4Hôpital Saint Louis Laboratoire de Parasitologie-Mycologie, Assistance Publique des Hôpitaux de Paris, 75010 Paris, France; 5Laboratoire Bioclinic, 75009 Paris, France; 6Service de Parasitologie-Mycologie, Hôpitaux Universitaires Henri Mondor, AP-HP, 94000 Créteil, France; 7UR Dynamic 7380, UPEC, EnvA, USC ANSES, Faculté de Santé, 94000 Créteil, France; 8Sorbonne Université, INSERM, CNRS, Centre d’Immunologie et des Maladies Infectieuses, Cimi-Paris, 75013 Paris, France; 9Service de Parasitologie-Mycologie, Hôpital Bichat-Claude Bernard, AP-HP, 75018 Paris, France; 10Univ Bordeaux, Centre de Recherche Cardio-Thoracique de Bordeaux, Inserm UMR 1045, 33600 Bordeaux, France; 11CHU Bordeaux, Département de Parasitologie-Mycologie, CIC 1401, 33000 Bordeaux, France; 12Service de Parasitologie-Mycologie, Hôpital Avicenne, AP-HP, 93009 Bobigny, France; 13Faculté de Médecine, Université Sorbonne Paris Nord, 93009 Bobigny, France; 14Sciensano, BCCM/IHEM Collection, Mycology and Aerobiology Unit, 1000 Brussels, Belgium; 15Institut Pierre Louis d’Épidémiologie et de Santé Publique, Inserm, Sorbonne Université, 75013 Paris, France

**Keywords:** MALDI-TOF mass spectrometry, *Trichophyton indotineae*, *Trichophyton mentagrophytes* species complex, MSI-2, fungi, dermatophyte

## Abstract

*Trichophyton indotineae* is an emerging pathogen which recently spread from India to Europe and that is more prone than other species of the *Trichophyton mentagrophytes* complex to show resistance to terbinafine, resulting in the necessity of rapid identification. Here, we improved the online MSI-2 MALDI-TOF identification tool in order to identify *T. indotineae*. By multiplying the culture conditions (2 culture media and 6 stages of growth) prior to protein extractions for both test isolates and reference strains, we added 142 references corresponding to 12 strains inside the *T. mentagrophytes* complex in the online MSI-2 database, of which 3 are *T. indotineae* strains. The resulting database was tested with 1566 spectra of 67 isolates from the *T. mentagrophytes* complex, including 16 *T. indotineae* isolates. Using the newly improved MSI-2 database, we increased the identification rate of *T. indotineae* from 5% to 96%, with a sensitivity of 99.6%. We also identified specific peaks (6834/6845 daltons and 10,634/10,680 daltons) allowing for the distinction of *T. indotineae* from the other species of the complex. Our improved version of the MSI-2 application allows for the identification of *T. indotineae*. This will improve the epidemiological knowledge of the spread of this species throughout the world and will help to improve patient care.

## 1. Introduction

Species belonging to the *Trichophyton mentagrophytes* complex (*T. mentagrophytes, T. interdigitale* and *T. indotineae*) present the same morphological characteristics in laboratory culture media and are thus difficult to distinguish by phenotypic methods [1]. From 2008 to 2016, the mycology community dropped the distinction between *Trichophyton interdigitale* and *T. mentagrophytes*, and the only distinction made was between the origin of the isolate, which was either anthropophilic or zoophilic [2]. In 2017, a revision of the dermatophyte taxonomy reinstated the distinction between the two species [3], and a work performed in 2019 classified different genotypes of *T. mentagrophytes* and *T. interdigitale* depending on the source of infection, the geographic origin, and the clinical signs [4]. To date, a total of 6 *T. interdigitale* and 22 *T. mentagrophytes* genotypes have been described [4,5], differing by 1 to 9 nucleotides in the ITS sequence. Genotypes of *T. interdigitale* correspond to anthropophilic genotypes incriminated in tinea pedis or tinea unguium lesions, whereas genotypes of *T. mentagrophytes* are zoophilic, and are responsible for inflammatory lesions in humans [3]. Since 2016, dermatologists have pointed out the emergence in India of extensive and difficult-to-treat dermatophytosis, especially tinea corporis and tinea cruris [6]. In 2018, Singh et al. [7] were the first to associate the phenomenon with a specific clade of strains inside the *T. mentagrophytes* complex (up to 75% of the isolates in India, according to Ebert et al. [8]) that were frequently resistant to terbinafine, and they associated the outbreak with *T. mentagrophytes* genotype VIII. Even though this genotype was associated with *T. mentagrophytes,* the zoophilic or geophilic origin was never proven, and Jabet et al. [9], by analyzing all *T. mentagrophytes* sequences available on NCBI, identified only a handful of cases on animals. In 2019, Nenoff et al. [10] pointed out the need for clarification of *T. mentagrophytes* species to avoid misinterpretation during clinical outbreaks. After some taxonomic controversies, a new species name, *T. indotineae*, was introduced for this clade based on specific molecular features [11,12]. The emergence of this particular species is supposedly associated with the intensive use of creams combining terbinafine and corticoids in India [13].

After its emergence in India, *T. indotineae* spread into many countries, with cases reported in Japan [11], Canada [14], and in European countries such as Germany [15], Switzerland [5], Greece [16], Denmark [17], and France [9,18]. Facing the impressive development of this dermatophytosis agent in India, and later on in other countries, mycologists proposed to include this disease in the list of emerging infectious diseases [1]. The spread of this new species makes it necessary to develop diagnostic tools capable of accurately differentiating it from the other species belonging to the *T. mentagrophytes* complex which are also responsible for tinea cruris and tinea corporis.

As *T. indotineae* is most phenotypically similar to *T. mentagrophytes* or *T. interdigitale*, the only available method to currently identify it is by molecular biology, either by sequencing of the ITS region or by using recently developed PCR-based assays [19]. This method, despite the development of the next generation sequencing technique that is becoming more and more affordable, remains the easiest method to use in a mycology laboratory, but has the disadvantages of being expensive and time consuming. Moreover, distinction between the different genotypes requires the alignment of the sequences for which the whole ITS region is necessary, but not always available. 

Currently, many mycology laboratories have access to MALDI-TOF mass spectrometers, allowing for the use of fast and accurate identification procedures for fungal species when coupled with appropriate references. Recently, an encouraging study by Tang et al. [20] showed that MALDI-TOF spectra belonging to *T. indotineae* could be distinguished from spectra of the other species of the *T. mentagrophytes* species complex, as they clustered together in a distant MALDI-TOF MS tree. Since 2017, online applications (MSI— Mass Spectrometry Identification) [21], MSI-2 [22]) have been made available free of charge for hospital laboratory use. MSI-2 is an application developed by Sorbonne University (Paris, France) and the Belgian collection of microorganisms (BCCM-IHEM, Brussels, Belgium) devoted to MALDI-TOF spectra identification, and using original identification algorithms and original reference databases (Fungi, *Leishmania*, Dipters, *Amanita*, *Phlebotomus,* and *Enterobacter cloacae*). The fungal MSI-2 database, of interest here, contains 16,442 references corresponding to 1581 fungal species. Even if MSI-2 has recently been improved for the identification of frequently encountered dermatophyte species [23], the identification of *T. indotineae* requires supplementary developments. After the database update by Jabet et al., it has been reported by MSI-2 users that the application was unable to identify *T. indotineae*, despite the presence of references from two collection strains in the database. Prior to this investigation, we tested the ability of the application to identify 76 spectra from 19 isolates of *T. indotineae* at day 5 of subculture on Sabouraud Chloramphenicol Gentamicin agar (SDA-CG) and 76 spectra from day 5 on Id-Fungi Plates (IDFP): we obtained only 8 and 23 spectra, respectively, that were correctly identified, confirming the observation of the MSI-2 users. 

Therefore, in this study, we established new references and assessed the ability of the resulting new spectra library (available at https://msi.happy-dev.fr/, accessed on October 2022) to distinguish *T. indotineae*, regardless of the culture medium or the time of growth, from other species of the *T. mentagrophytes* complex, especially genotypes that are the most frequently occurring in France and Europe, i.e., genotypes I, II, II*, III*, VI and VII [5].

## 2. Materials and Methods

### 2.1. Database

In this study, an improved version of the MSI-2 database was built by adding new spectra to the online MSI-2 library (called MSI-2-improved throughout the manuscript). These new reference spectra were obtained at various stages of growth (days 3, 5, 7, 10, 12, and 14) and on two culture media—IDFP (Conidia, Lyon, France) and SDA-CG (Oxoid, France). 

### 2.2. Panels of Strains and Isolates

The capacity of the improved MSI-2 database to identify *Trichophyton indotineae* was tested with two different panels to assess both sensitivity and specificity.

Panel 1 is composed of 2879 spectra corresponding to 67 isolates belonging to the *T. mentagrophytes* species complex that have been obtained from cultures of human samples in several French mycology laboratories, from the Sciensano collection and from animal samples. The list and origin of the spectra constituting Panel 1 can be found in Appendix A. Among the 67 isolates, 1313 spectra from 34 strains were obtained by Jabet et al. in 2022 [23], under the same culture conditions as the ones described above. Regarding those 34 strains that are present in both the MSI-2-improved database and Panel 1, self-recognition of spectra belonging to the same strain were discarded in order to obtain only the best identification with a different reference strain.

Panel 2 is an outgroup of 5212 spectra from various fungal species (*Aspergillus*, *Fusarium*, filamentous fungi from daily workflows of clinical laboratories, and dermatophytes outside the *T. mentagrophytes* complex) that we have already used to test the specificity of several databases. Its composition has previously been published [21,22,23,24,25].

### 2.3. Trichophyton mentagrophytes Genotyping

All *T. mentagrophytes* isolates were typed by sequencing the ITS region (ITS1: 5′-TCC GTA GGT GAA CCT GCG G-3′; ITS4: 5′-TCC TCC GCT TAT TGA TAT GC-3′) [26], with an annealing temperature of 55 °C. Distinctions between the genotypes were made by comparison with the reference sequences published by Taghipour et al. [4] and by Klinger et al. [5] (accession numbers of the reference sequences can be found in Appendix A). Sequences were aligned using MEGA X software, and genotypes were distinguished by comparison of the nucleotides one by one in order to distinguish genotypes that differ by one deletion or insertion (as is the case between genotype I and genotype II). Typing results were verified with the Python script (https://github.com/Ivan-Pchelin/genotyping-by-sequencing, accessed on 7 October 2022) developed by Ivan-Pchelin in 2019 [4]. Genotyping of the 67 isolates/strains constituting Panel 1 was performed by alignment of the full ITS sequences obtained in our laboratory. Genotyping of the MSI-2 reference strains was also checked by aligning sequences obtained by Jabet et al. [23] and by the Sciensano laboratory at the time of adding strains to their collection.

### 2.4. MALDI-TOF Mass Spectra

All spectra used for Panel 1 were acquired from cultures on both SDA-CG agar plates and IDFP agar plates at various stages of growth between 1 and 21 days of culture. The cultures were sampled with a sterile scalpel blade and introduced into a sterile microtube containing 70% ethanol. A full extraction method consisting of an inactivation step in 70% ethanol, a lysis step with 70% formic acid, and a protein precipitation step with 100% acetonitrile was performed in all cases, as previously published [21]. Protein extracts were deposited in successive quadruplicates, as proposed in previous MALDI-TOF MS studies on filamentous fungi and dermatophytes [27,28]. The identification obtained with the highest score for the replicate harvesting was retained for the assessment of both databases. The different steps of the study are presented in the study design (Figure 1).

### 2.5. Method of Spectral Comparison

All spectra of *T. indotineae* obtained for this study were compared to observe modifications of the spectra related to the stage of the culture. To visualize the differences, we used ClinProTool software (Bruker Daltonics, version 3.0). Spectra corresponding to the six stages of extraction per culture medium were loaded as six independent groups of spectra. The standard preparation workflow of the software, consisting of baseline subtraction and normalization, was applied using the default parameters. Then, a recalibration, consisting of reducing the mass shifts that may occur during the measurement of the spectra, was performed. Spectra were visualized in Gel/Stack View mode.

### 2.6. Peak Picking and Distinction between the Two Most Represented Genotypes, i.e., T. interdigitale and T. indotineae

As the two groups represented for all stages of growth by a sufficient number of spectra are *T. interdigitale* genotype II (1108 spectra) and *T. indotineae* (905 spectra), we compared the spectra of those two species to search for specific peaks after a pretreatment step of the spectra, consisting of baseline subtraction, smoothing, alignment, and normalization of the intensities. Boxplots were then computed to compare peak intensities in relation to the species (*T. interdigitale* genotype II versus *T. indotineae*).

Finally, we extracted the list of peaks present in all of the spectra obtained at days 3, 5, 7, 10, 12, and 14 using Flex Analysis software, and we calculated the percentage of spectra that contained the species-specific peaks (*T. indotineae* vs. *T. interdigitale/T. mentagrophytes*), as well as the per day of growth, to assess the specificity of the peaks.

## 3. Results

### 3.1. Trichophyton mentagrophytes Genotyping

Control of the genotypes present in the MSI-2 database using their ITS sequences shows the presence of 8 genotypes among the references (*T. interdigitale* genotypes I and II; *T. mentagrophytes* genotypes II*, III, III*, IV, and XXIV; *T. indotineae*), represented by 1 to 25 strains (Appendix A). Among the 67 isolates constituting panel 1, we obtained 8 genotypes: *T. interdigitale* genotype I (n = 8), *T. interdigitale* genotype II (n = 26), *T. mentagrophytes* genotype II* (n = 3), *T. mentagrophytes* genotype III (n = 1), *T. mentagrophytes* genotype III* (n = 8), *T. mentagrophytes* genotype VI (n = 1), *T. mentagrophytes* genotype VII (n = 1), and *T. indotineae* (n = 19).

### 3.2. MSI-2 Improved Database

A total of 142 new reference spectra were added to the database previously published by Jabet in 2022. New references represented not only *T. indotineae* (n = 3), but also *T. interdigitale* genotypes I (n = 2 strains) and II (n= 3 strains) and *T. mentagrophytes* genotypes II*, III*, VI, and VII (n = 1 strain each). To build the MSI-2-improved database, *T. interdigitale* and *T. indotineae* strains were randomly selected from all the strains that were available for this study. A list of the references of the *T. mentagrophytes* species complex can be found in Appendix A.

### 3.3. Identification Performances Using the MSI-2-Improved Database

Using the MSI-2-improved database, we built a contingency table for all culture conditions combined. The identification results per isolate and per culture condition (n = 722) are computed in Table 1. The number of spectra, number of true and false positive and negative identifications, sensitivity, and specificity for the three species are compiled in Table 2.

While only 12.12% of the *T. indotineae* spectra acquired before 5 days of growth and 5.31% of spectra acquired at the various stages of growth were correctly identified as *T. indotineae* with the MSI-2 database as published by Jabet et al. [23], using the improved database, the percentage of correct identifications of *T. indotineae* spectra was 96.02% overall (91.25% for extractions performed before day 5 of growth, 98.03% for extractions performed between day 6 and day 10, and 97.37% for extractions performed after day 11), while distinction between *T. interdigitale* and *T. mentagrophytes* remains imperfect with 81% of *T. interdigitale* isolates and 68% of *T. mentagrophytes*, respectively, identified as such.

### 3.4. Spectra Evolution over Time

The spectra of *T. indotineae* tended to change according to the stage of the culture, with some major peaks in the spectra from early culture tending to decrease in intensity over time (the most important peaks in both IDFP and SDA-CG media being the peaks at 3056, 6115, and 6845 daltons), while others increased in intensity (such as the triplex set of peaks at 5342, 5328, 5399, or the pair of peaks at 10,666 and 10,676 daltons), as shown in Figure 2A.

Overall, there is no apparition or disparition of peaks with the aging of the colony, the only alteration being the intensity of some peaks. Regarding the MSI-2 identification process, this might be of importance, as the peaks are classified depending on their intesity for the identification process. Hence, a reference corresponding to 14 days of growth would not have the same ordering of the masses corresponding to the various peaks as a reference created at day 3 of growth.

### 3.5. Detection of Specific Peaks

We highlighted the importance of two sets of peaks that allow the distinction between *T. indotineae* and the tested genotypes of the *T. mentagrophytes* species complex. Boxplots showing the distribution of intensities for these more discriminant peaks, allowing for *T. indotineae* identification among the *T. mentagrophytes* complex, are presented in Figure 3.

Using the peak list obtained with Flex Analysis software, we calculated the percentage of spectra from Panel 1 for each genotype that contained these four peaks. The results are found in Table 3.

The presence/absence of the peaks at 6834/6845 daltons determined between 7 and 10 days of growth seem to be the best discriminating peaks for the distinction between *T. indotineae* and *T. interdigitale/mentagrophytes*, as 100% of the tested *T. indotineae* spectra contain the 6845 Da peak, and 0% contain the 6834 Da peak. Considering the combination of the peaks at 6834 and 10,634 (for the non-*T. indotineae* species) or 6845 and 10,680 (for *T. indotineae*) (i.e., when both discriminating peaks are present in one spectrum) prevents the misidentifications obtained for the non-*T. indotineae* species when considering only the peak at 6845 daltons, or the misidentifications of *T. indotineae* when considering only the peak at 10,634 daltons.

## 4. Discussion

*T. indotineae* was formerly known as *T. mentagrophytes* genotype VIII. Its recent rise to the rank of a species, as discussed by Kano et al. in 2020 [11], and the fact that *T. indotineae* is more prone to be resistant to terbinafine increase the need for simple identification tools that can distinguish it from the other species of the *T. mentagrophytes* complex. To date, mycologists have had to resort to DNA sequencing to identify this species.

Prior to recent investigations performed on the Indian epidemics, resistance to terbinafine among dermatophytes was out of the ordinary. Moreover, as *Trichophyton* species are slow-growing pathogens on agar media, techniques of resistance investigation are not usually performed with the intention of evaluating whether terbinafine treatment would be effective. In 2014, the British Association of Dermatologists (BAD) recommended treating dermatophytoses, especially onychomycosis, with terbinafine [29]. In 2020, Rengasamy and the Indian Association of Dermatologists, Venereologists, and Leprologists (IADVL) Task Force against Recalcitrant Tinea decided to use terbinafine as a first-line systemic agent in the treatment of terbinafine naïve patients with glabrous tinea [30]. Hence, today, physicians tend to prescribe only this one molecule as a first line of treatment when there is a need to give an oral treatment for dermatophyte infections. Studies have recently shown that the drug of choice for the treatment of infections by *T. indotineae* would be itraconazole [31], hence the necessity to rapidly identify this pathogen to increase the chances of healing. This study shows that differentiating *T. indotineae* from *T. interdigitale* or *T. mentagrophytes* is possible using MALDI-TOF MS tools by using an appropriate spectral database and taking into account the evolution of spectra according to the stage of the fungal colonies.

Even though *T. indotineae* is known to more often be resistant to treatment with terbinafine than other *Trichophyton* species, not all *T. indotineae* isolates are resistant to treatment. In the literature, resistance to terbinafine was observed for 16 [32] to 76% [8] of the isolates studied. In our study, we did not try to achieve resistance identification through MALDI-TOF mass spectrometry. This resistance is linked to several mutations in the squalene epoxidase (SQLE) gene [33,34,35], and the SQLE protein is a relatively large one (55 kDa for *Saccharomyces cerevisiae*; https://www.uniprot.org/uniprotkb/P32476/entry, accessed on 28 September 2022) that cannot be detected using the universal MALDI-TOF identification protocol for microorganisms (2–20 kDa). Detecting the SQLE protein and its mutations through MALDI-TOF MS requires different Microflex parameters and a different methodology than the one applied in microbiology laboratories. This is why we decided here to prove that we could identify this new species using MALDI-TOF MS and an appropriate MS database as a first identification step in the process of treating a patient with tinea corporis or tinea cruris.

The MSI-2 database published by Jabet et al. in 2021 [23] was not initially tested with *T. indotineae* isolates, but was reported to yield poor performance regarding the identification of *T. indotineae*. As in the article by Jabet et al., we decided to work with SDA-CG and IDFP media to both improve the MSI-2 database and test the identification process. SDA-CG is the culture medium that is most often used in microbiology laboratories to grow filamentous fungi and dermatophytes. It is sometimes supplemented with cycloheximide to favor the growth of dermatophytes, compared to fast growing fungi such as *Aspergillus* spp, but to the best of our knowledge, cycloheximide does not interfere with the peaks present in the mass spectrum of a dermatophyte [27]. Recently, the IDFP culture medium has been introduced in the mycology landscape as a medium of choice to use as a sub-culture medium when identifying poorly sporulating fungi that are difficult to sample without collecting culture medium. Indeed, this medium presents a porous membrane on its surface on which the fungus can grow, allowing for easy sampling without danger of collecting agar, whatever the sampling tool chosen. Its use in mycology laboratories is rapidly increasing, especially for dermatophytes, as a spectrum can be obtained as soon as 48 h after sub-culturing. Other culture media such as Potatoes Dextrose Agar or Malt agar can be used in the search for dermatophytoses agents, but the use of those culture media remain occasional, and we focused on the most commonly used media to improve the reference database. In fine, sensitivities for *T. indotineae* (95.6%), *T. interdigitale* (81.2%), and *T. mentagrophytes* (67.9%) are high with the improved MSI-2 database. Improvement of the MSI-2 database also led to a high specificity for the three species, with only eight spectra corresponding to two strains of *T. interdigitale* and one strain of *T. mentagrophytes* misidentified as *T. indotineae*.

Using MALDI-TOF mass spectrometry to rapidly identify *T*. *indotineae* can lead biologists to test the targeted isolates for antifungal susceptibility to help physicians find better treatments for patients, improving the monitoring of the spread and epidemiology of this newly described species. When doubt remains with respect to the identification of *T. indotineae* and other species of the *T. mentagrophytes* species complex, biologists can look for the two peaks that have been found to be specific to *T. indotineae* (6845 Da and 10,680 Da), especially if the spectra were obtained after 7 days of growth.

One limitation of the current study is that we could not test all genotypes that have been identified thus far. Indeed, *T. indotineae*, *T. interdigitale* genotype II, and *T. mentagrophytes* genotype III* accounted for 82% of our isolates, as has been shown in other studies [4,5]. This study focused on genotypes that were available in the participating laboratories and in the Belgian collection of microorganisms that centralizes isolates from all around the world. Several false-positive identifications of spectra were found in the identification of *T. indotineae* in our panel of strains, but none of them corresponded to the highest score among the four replicates. Hence, there was no false-positive when taking into account the best result of the four replicates. It would be of interest to identify the protein corresponding to the two specific peaks to develop a rapid diagnostic test for laboratories that do not use MALDI-TOF MS or for field studies.

As it has been done in this study for *T. indotineae*, it would be of great interest to try to distinguish other genotypes of the *T. mentagrophytes* species complex, as some of them are responsible for particular cutaneous afflictions. For example, this is the case for *T. mentagrophytes* genotype VII that causes an inflammatory tinea genitalis affliction [5]. Such experimentations requires the availability of a large number of strains from the same genotype, which are currently not easy to come by.

## 5. Conclusions

In conclusion, the high sensitivity and specificity obtained for the identification of *T. indotineae* using the online MSI-2 application will allow MALDI-TOF users to rapidly identify *T. indotineae* from a positive culture of dermatophytes with this database. This will also improve diagnosis and help to monitor the spread of this species around the world, as laboratories proceeding to their identification through MSI-2 use the same database and are connected through the network of users. By more easily identifying *T. indotineae* isolates, biologists will more successfully target isolates that are potentially resistant to terbinafine and will trigger the susceptibility testing of those isolates more efficiently through the results obtained by MALDI-TOF MS.

## Figures and Tables

**Figure 1 jof-08-01103-f001:**
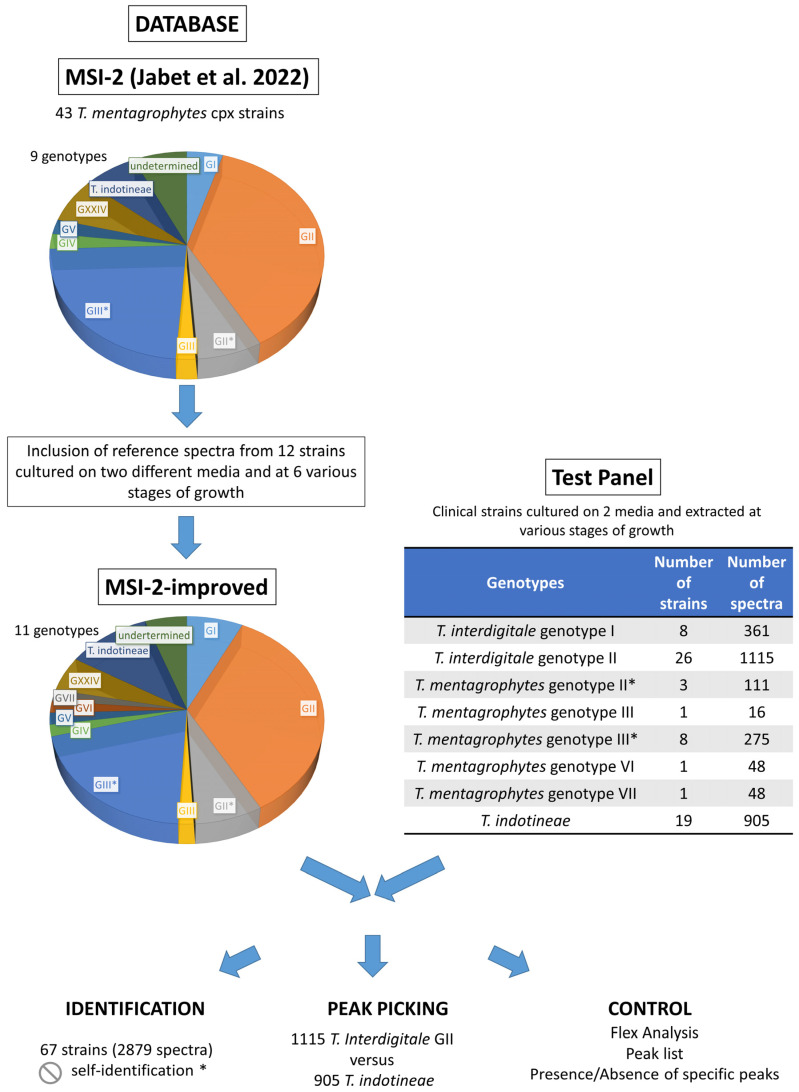
Study design [23]. * For isolates that are in both the reference database and in the test panel, self-identification at the strain level is not allowed.

**Figure 2 jof-08-01103-f002:**
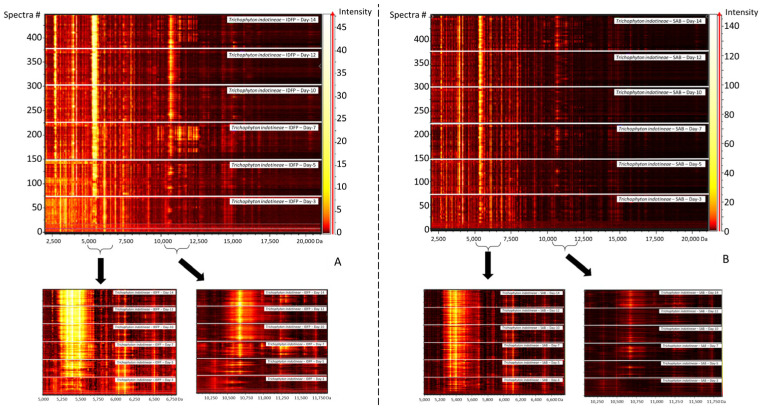
Comparison of the aligned spectra obtained at the various stages of growth for *T. indotineae* isolates on both IDFP (**A**) and SDA-CG agar (**B**). To visualize increases or decreases in the intensities of some peaks, a zoomed-in view of two portions of the spectra (5000–16750 Da; 10,000–11,750 Da) is shown. The highest peaks in intensity are represented by the brightest colors.

**Figure 3 jof-08-01103-f003:**
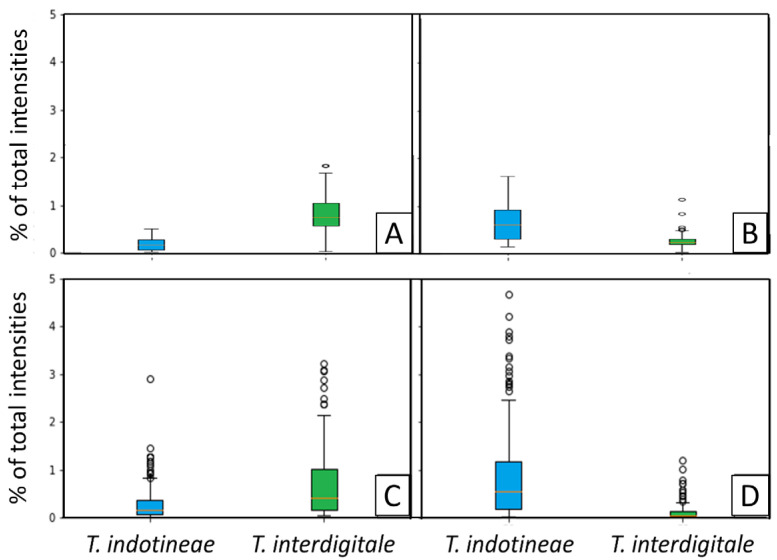
Distribution of the percentages of the intensities corresponding to one specific peak compared to the overall intensities of the spectrum at 10 days of growth. *T. indotineae* plots are colored blue, and *T. interdigitale* genotype II plots are colored green. (**A**) = peak at 6834 (±5) daltons (Da), (**B**) = peak at 6845 (±5) Da, (**C**) = peak at 10,634 (±10) Da, and (**D**) = peak at 10,680 (±10) Da.

**Table 1 jof-08-01103-t001:** Number of isolates of Panel 1 (best identification among the four replicates per culture condition) identified as *T. indotineae*, *T. interdigitale*, *T. mentagrophytes*, other species, or under the score threshold of 20 for the MSI-2-improved database.

	Identification by the MSI-2-Improved Database				
Species	*T. indotineae*	*T. interdigitale*	*T. mentagrophytes*	Other	Under Threshold
*T. indotineae* (n = 226)	217	3	4	2	0
*T. interdigitale* (n = 371)	0	301	64	6	0
*T. mentagrophytes* (n = 125)	0	37	85	3	0

**Table 2 jof-08-01103-t002:** Number of spectra, sensitivity, specificity, and number of true positive, true negative, false positive, and false negative identifications for the three species.

Species	Number of Spectra	True Positive	True Negative	False Positive	False Negative	Sensitivity	Specificity
*T. indotineae*	905	865	1966	8	40	95.6%	99.6%
*T. interdigitale*	1476	1198	1252	151	278	81.2%	89.2%
*T. mentagrophytes*	498	338	2125	256	160	67.9%	89.2%

Regarding the 5212 spectra constituting the outgroup, there was no misidentification as *T. indotineae*. The only spectra that matched against references belonging to the *T. mentagrophytes* species complex were 16 spectra of Trichophyton tonsurans that were misidentified as *T. interdigitale* (n = 11) and *T. mentagrophytes* (n = 5).

**Table 3 jof-08-01103-t003:** Percentage of spectra containing the specific peak in the *T. mentagrophytes/interdigitale* spectra and in the *T. indotineae* spectra.

Peak (Da)	Species	J3%	J5%	J7%	J10%	J12%	J14%
6834	*T. indotineae*	7	5	0	0	0	0
	others	98	100	100	100	100	99
6845	*T. indotineae*	97	92	100	99	88	82
	others	65	5	18	10	33	31
10,634	*T. indotineae*	5	8	11	11	3	9
	others	19	68	81	89	94	90
10,680	*T. indotineae*	19	63	92	91	93	97
	others	3	2	0	2	2	9
6834 and 10,634	*T. indotineae*	0	1	0	0	0	0
	others	19	68	81	89	94	89
6845 and 10,680	*T. indotineae*	19	55	92	91	82	79
	others	0	0	0	0	0	3

## Data Availability

All data can be access by writing to the corresponding author.

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
