# Peer review of "MALDI-TOF Mass Spectrometry Online Identification of Trichophyton indotineae Using the MSI-2 Application"

_jof, 2022, doi:10.3390/jof8101103_

Round 1

Reviewer 1 Report

Line 36, 42. The Abstract may seem confusing. What was the primary result of the study? Was it a development of a tool or rather an improvement to the database?

Lines 42, 45. Is MSI-2 a database or an application?

Line 47. “To” is missing.

Lines 87-88. Please consider mentioning recently developed PCR-based assays for T. indotineae identification [1].

Lines 93-94. “most mycology laboratories are equipped with MALDI TOF mass spectrometers”. Please provide some reference to the statistics or maybe soften the expression.

Lines 93-108. The part of the Introduction devoted to MALDI-TOF MS may deserve its own paragraph. Also, I would request better explanation of what is MSI-2 and how it is used.

Line 109. May “therefore” be a better choice than “hence”?

Line 113 and further in the text. Instead of concurrent use of different versions of genotype nomenclature, please choose either form: “T. mentagrophytes / T. interdigitale genotype I” or “T. mentagrophytes / T. interdigitale Type I”, and use consistently.

Line 114. Please cite the literature source of the primers.

Line 116. “A new MSI-2 database”. Probably, not new, but improved one or supplemented with new mass spectra or the like.

Lines 148-150. The method seems to be excessively laborious, since there is publicly available Python script for molecular typing of T. mentagrophytes / T. interdigitale species complex isolates [2].

Lines 156-157. Culture media manufacturer names are not needed, since they were mentioned earlier.

Line 169. Please disclose the version of ClinProTool software.

Lines 169-170. “Independent groups”. To my perception, the phrase is not clear enough. The groups of what?

Line 174. “Gel/Stack View”. Maybe, “Gel/Stack View mode” is better?

Lines 175-185. From the section title, it is unclear why the distinction was made only between T. interdigitale and T. indotineae. Why T. mentagrophytes was not included? Why boxplot analysis was done only for T. mentagrophytes and T. indotineae? Please revise the paragraph. Please also decide whether additional calculations are needed.

Line 184. “the peaks of interest per species”. Species-specific peaks?

Lines 223 and 228. Does “Sabouraud media” (“medium”?) differ from “Sabouraud GC agar”? Please use consistent medium names throughout the text.

Line 274. “physicians tend to prescribe only one molecule (terbinafine) for dermatophyte infections”. Please do cite a couple of relevant clinical sources. A correction may be needed.

Lines 282-283. “all T. indotineae isolates are not systematically resistant”. “not all T. indotineae isolates are resistant”?

Line 284. “we did not tried”. “We did not try”?

Line 285. In fact, a number of mutations are known [3,4].

Lines 347-348. The word “users” appears in the same sentence twice. Please consider an amendment.

Lines 349-350. Your methodological improvements are not directly related to susceptibility testing. Please revise the phrase.

1.                  Batvandi A, Pchelin IM, Kiasat N, Kharazi M, Mohammadi R, Zomorodian K, Rezaei-Matehkolaei A. Time and cost-efficient identification of Trichophyton indotineae. Mycoses. 2022. doi: 10.1111/myc.13530.

2.                  Taghipour S, Pchelin IM, Zarei Mahmoudabadi A, Ansari S, Katiraee F, Rafiei A, Shokohi T, Abastabar M, Taraskina AE, Kermani F, Diba K, Nouripour-Sisakht S, Najafzadeh MJ, Pakshir K, Zomorodian K, Ahmadikia K, Rezaei-Matehkolaei A. Trichophyton mentagrophytes and T interdigitale genotypes are associated with particular geographic areas and clinical manifestations. Mycoses. 2019;62(11):1084-1091. doi: 10.1111/myc.12993.

3.                  Yamada T, Maeda M, Alshahni MM, Tanaka R, Yaguchi T, Bontems O, Salamin K, Fratti M, Monod M. Terbinafine Resistance of Trichophyton Clinical Isolates Caused by Specific Point Mutations in the Squalene Epoxidase Gene. Antimicrob Agents Chemother. 2017;61(7):e00115-17. doi: 10.1128/AAC.00115-17.

4.                  Burmester A, Hipler UC, Elsner P, Wiegand C. Point mutations in the squalene epoxidase erg1 and sterol 14-α demethylase erg11 gene of T indotineae isolates indicate that the resistant mutant strains evolved independently. Mycoses. 2022;65(1):97-102. doi: 10.1111/myc.13393.

Author Response

Reviewer 1

Line 36, 42. The Abstract may seem confusing. What was the primary result of the study? Was it a development of a tool or rather an improvement to the database?

Thank you for pointing out that the abstract lacked clearness. It has been modified accordingly: we now state that the database was improved in order to allow the identification of T. indotineae. “Here, we improved the online MSI-2 MALDI- TOF identification tool in order to identify T. indotineae.”

Lines 42, 45. Is MSI-2 a database or an application?

We understand the reviewer confusion as MSI-2 is both an application that is available online, with original identification algorithms, and a database linked to it. We now explain it in the text as asked a few point later.

Line 47. “To” is missing.

Sorry for the mistake. It has been added to the text.

Lines 87-88. Please consider mentioning recently developed PCR-based assays for T. indotineae identification [1].

At first, we did not think that mentioning PCR-Assays was necessary for the introduction on MALDI TOF, but after reading the reviewer comment, we agree that it was lacking from this part of the introduction. So we added a sentence on PCR-based assays, with the reference mentioned by the reviewer: “As T. indotineae is mostly phenotypically similar to T. mentagrophytes or T. interdigitale, the only available method to currently identify it is by molecular biology, either by sequencing of the ITS region or with recently developed PCR-based assays [19]”

Lines 93-94. “most mycology laboratories are equipped with MALDI TOF mass spectrometers”. Please provide some reference to the statistics or maybe soften the expression.

We were wrong to make this statement and we agree with the reviewer on the matter. To the best of our knowledge, Mycology departments may not be directly equipped with MALDI TOF machines, but at least, they share it with the bacteriology laboratory, or have an access to it. We modified our sentence to be less categorical in our statement: “Currently, a lot of mycology laboratories have access to MALDI- TOF mass spectrometers…”

Lines 93-108. The part of the Introduction devoted to MALDI-TOF MS may deserve its own paragraph. Also, I would request better explanation of what is MSI-2 and how it is used.

Thank you to the reviewer for pointing out that not all reader know about the MSI-2 application. We added a description of the MSI-2 application and database there in the text, in a dedicated paragraph: “Since 2017, online applications [MSI (for Mass-Spectrometry Identification) [21], MSI-2 [22]] have been made available free of charge for hospital laboratory use. MSI-2 is an application developed by Sorbonne University (Paris, France) and the Belgian collection of microorganisms (BCCM-IHEM, Brussels, Belgium) devoted to MALDI-TOF spectra identification, and using original identification algorithms and original reference databases (Fungi, Leishmania, Dipters, Amanita, Phlebotomus and Enterobacter cloacae). The fungal MSI-2 database, of interest here, contains 16,442 references corresponding to 1,581 fungal species (accessed on October 2022).”

Line 109. May “therefore” be a better choice than “hence”?

We agree with the reviewer and replaced “Hence” by “Therefore” in this sentence.

Line 113 and further in the text. Instead of concurrent use of different versions of genotype nomenclature, please choose either form: “T. mentagrophytes T. interdigitale genotype I” or “T. mentagrophytes T. interdigitale Type I”, and use consistently.

Thank you for pointing out the lack of consistency in our nomenclature for genotypes. We decided to use “genotype X” and replaced all other terminologies throughout the text.

Line 114. Please cite the literature source of the primers.

Sorry for overlooking this reference. We introduced the article by the ISHAM society, written by Irinyi and listing many ITS primers including the two that we used.

Line 116. “A new MSI-2 database”. Probably, not new, but improved one or supplemented with new mass spectra or the like.

We modified this sentence accordingly

Lines 148-150. The method seems to be excessively laborious, since there is publicly available Python script for molecular typing of T. mentagrophytes T. interdigitale species complex isolates [2].

We thank the reviewer for letting us know about this python script that we overlooked. I tested it and found the same identification results than with the laborious technique that we used. Hence, instead of removing the description of our process, we added a sentence including the genotyping script by Ivan-Pchelin: “Typing results were verified with the python script (https://github.com/Ivan-Pchelin/genotyping-by-sequencing) developed by Ivan-Pchelin in 2019 [4].”

Lines 156-157. Culture media manufacturer names are not needed, since they were mentioned earlier.

Sorry for the redundancy. We removed the names of the manufacturers from this section of the manuscript.

Line 169. Please disclose the version of ClinProTool software.

We used version version 3.0, and it is now mentioned in the manuscript.

Lines 169-170. “Independent groups”. To my perception, the phrase is not clear enough. The groups of what?

Sorry for the fogginess of this sentence. We amended it so that the reader can understand that these are groups of spectra: “Spectra corresponding to the six stages of extraction per culture medium were loaded as six independent groups of spectra.”

Line 174. “Gel/Stack View”. Maybe, “Gel/Stack View mode” is better?

We modified the text accordingly

Lines 175-185. From the section title, it is unclear why the distinction was made only between T. interdigitale and T. indotineae. Why T. mentagrophytes was not included?

We agree with the reviewer that this is a sensible point to explain and to understand for the reader. In fact, spectra were acquired during two periods. The first period, corresponding to the spectra obtained by Jabet, and during which there was no precise timing for the protein extraction. The protocol was to obtain 5 stages of growth for the cultures on Sabouraud and 3 stages of growth on IDFP. During this first period, a partnership was established with the veterinary school of Maison Alfort (France), and many T. mentagrophytes were obtained. The second period of acquisition regarded the T. indotineae isolates and prospective T. interdigitale/mentagrophytes isolates obtained in the participating mycology laboratories. For this second period, it was decided to obtain spectra at precise stages of growth, i.e. days 3, 5, 7, 10, 12, and 14. During this second period, we obtained T. interdigital genotype II in majority, and very few T. mentagrophytes. As most of the T. mentagrophytes genotypes were obtained during the first period of spectra collection, the stages of growth were not always compatible with the stages obtained for T. indotineae during the second period. Therefore, we decided to compare the two most represented genotypes that were obtained with the same protocol (T. interdigitale genotype II and T. indotineae). We modified the title of the section so that the reason why we chose those two genotypes is more clear: “Peak picking and distinction between the two most represented genotypes, i.e. T. interdigitale and T. indotineae

Why boxplot analysis was done only for T. mentagrophytes and T. indotineae? Please revise the paragraph. Please also decide whether additional calculations are needed.

We thank the reviewer for his/her careful reading of the paper that highlighted this mistake. Indeed, boxplots were obtained for T. interdigitale genotype II and T. indotineae. The correction has been made in the text.

Line 184. “the peaks of interest per species”. Species-specific peaks?

The text was modified accordingly

Lines 223 and 228. Does “Sabouraud media” (“medium”?) differ from “Sabouraud GC agar”? Please use consistent medium names throughout the text.

Once again, we are sorry for our lack of consistency in our nomenclature. We used only Sabouraud CG agar in this study. We modified all mentions of Sabouraud into SDA-CG throughout the manuscript.

Line 274. “physicians tend to prescribe only one molecule (terbinafine) for dermatophyte infections”. Please do cite a couple of relevant clinical sources. A correction may be needed.

As requested by the reviewer, we added two references regarding the guidelines of treatment for dermatophytoses, and amended our text regarding the tendencies of physicians to prescribe only one molecule: “The British Association of Dermatologists (BAD) recommended, in 2014, to treat derma-tophytoses, especially onychomycosis, with terbinafine [29]. In 2020, Rengasamy and the Indian association of Dermatologists, Venereologists and Leprologists (IADVL) Task Force against Recalcitrant Tinea decided to use terbinafine as a first line systemic agent in treatment of terbinafine naïve patients with glabrous tinea [30]. Hence, today, physicians tend to prescribe only this one molecule as first line of treatment when there is a need to give an oral treatment for dermatophyte infections.”

Lines 282-283. “all T. indotineae isolates are not systematically resistant”. “not all T. indotineae isolates are resistant”?

We modified the manuscript accordingly.

Line 284. “we did not tried”. “We did not try”?

The correction has been made.

Line 285. In fact, a number of mutations are known [3,4].

The reviewer is right, and we ourselves observed several mutations when testing our isolates. We added the word “several” and the reference by Burmester.

Lines 347-348. The word “users” appears in the same sentence twice. Please consider an amendment.

We reformulated the sentence so that the word “users” appear less often: “This will also improve diagnosis and help to monitor the spread of this species all around the world, as laboratories proceeding to their identification through MSI-2 use the same database and are connected through the network of users.”

Lines 349-350. Your methodological improvements are not directly related to susceptibility testing. Please revise the phrase.

Thank you to the reviewer for pointing out that we badly worded our sentence and that it could be confusing to the reader. We modified the last sentence so that the reader does not believe that we can predict the susceptibility of the isolated using our improved database: “By identifying more easily T. indotineae isolates, biologists will more easily target isolates that are potentially resistant to terbinafine and will trigger the susceptibility testing of those isolates more efficiently through the results obtained by MALDI- TOF MS.”

  1. Batvandi A, Pchelin IM, Kiasat N, Kharazi M, Mohammadi R, Zomorodian K, Rezaei-Matehkolaei A. Time and cost-efficient identification of Trichophyton indotineae. Mycoses. 2022. doi: 10.1111/myc.13530.
  2. Taghipour S, Pchelin IM, Zarei Mahmoudabadi A, Ansari S, Katiraee F, Rafiei A, Shokohi T, Abastabar M, Taraskina AE, Kermani F, Diba K, Nouripour-Sisakht S, Najafzadeh MJ, Pakshir K, Zomorodian K, Ahmadikia K, Rezaei-Matehkolaei A. Trichophyton mentagrophytes and T interdigitale genotypes are associated with particular geographic areas and clinical manifestations. Mycoses. 2019;62(11):1084-1091. doi: 10.1111/myc.12993.
  3. Yamada T, Maeda M, Alshahni MM, Tanaka R, Yaguchi T, Bontems O, Salamin K, Fratti M, Monod M. Terbinafine Resistance of Trichophyton Clinical Isolates Caused by Specific Point Mutations in the Squalene Epoxidase Gene. Antimicrob Agents Chemother. 2017;61(7):e00115-17. doi: 10.1128/AAC.00115-17.
  4. Burmester A, Hipler UC, Elsner P, Wiegand C. Point mutations in the squalene epoxidase erg1 and sterol 14-αdemethylase erg11 gene of T indotineae isolates indicate that the resistant mutant strains evolved independently. Mycoses. 2022;65(1):97-102. doi: 10.1111/myc.13393.

Reviewer 2 Report

Submitted to Jof by Normand et al (Sep 2022)

The Trichophyton mentagrophytes complex within the dermatophytes contains two anthropophilic species, Trichophyton interdigitale and Trichophyton indotineae, which cannot be differentiated from zoophilic strains in culture. T. indotineae is an emerging pathogen with the vast majority of strains resistant to terbinafine. This species was first isolated in India and has since spread worldwide. The identification of the species of the T. mentagrophytes complex can be done by ITS sequencing. In contrast, the existing MALDI TOF database was not sufficient to distinguish this species from zoophilic strains of T. mentagrophytes and T. interdigitale. Therefore, the authors completed a database (MS-2) for the identification of this species. Two peaks (6,845 Dal-43 10,680 Daltons) allow the distinction of T. indotineae with a sensitivity of 99. %.

Comments, remarks

The experimental work is well done, and the result is of interest. However, the result section should not focus solely on the testing of the MS-2 Improved database. MS-2 Improved was constructed in this study and is therefore part of the results obtained. Therefore, the content of lines 115-141 should be in the results section between 3.1 and 3.2, which requires a slight modification of the manuscript.

Authors should carefully check the numbering of the references in the text. It appears that reference [18] should be reference [22] (Jabet et al.) on line 295.

Author Response

Reviewer 2

The Trichophyton mentagrophytes complex within the dermatophytes contains two anthropophilic species, Trichophyton interdigitale and Trichophyton indotineae, which cannot be differentiated from zoophilic strains in culture. T. indotineae is an emerging pathogen with the vast majority of strains resistant to terbinafine. This species was first isolated in India and has since spread worldwide. The identification of the species of the T. mentagrophytes complex can be done by ITS sequencing. In contrast, the existing MALDI TOF database was not sufficient to distinguish this species from zoophilic strains of T. mentagrophytes and T. interdigitale. Therefore, the authors completed a database (MS-2) for the identification of this species. Two peaks (6,845 Dal-43 10,680 Daltons) allow the distinction of T. indotineae with a sensitivity of 99. %.

Comments, remarks

The experimental work is well done, and the result is of interest. However, the result section should not focus solely on the testing of the MS-2 Improved database. MS-2 Improved was constructed in this study and is therefore part of the results obtained. Therefore, the content of lines 115-141 should be in the results section between 3.1 and 3.2, which requires a slight modification of the manuscript.

We thank the reviewer for pointing out that the improved database is not part of the methods, but a result in itself. We removed the description from the methods, and added a paragraph in the result section: “3.2. MSI-2 improved database: One hundred and fourty two new reference spectra were added to the database already published by Jabet in 2022. New references represented not only T. indotineae (n=3), but also T. interdigitale genotypes I (n=2 strains) and II (n= 3 strains) and T. mentagrophytes genotypes II*, III*, VI and VII (n=1 strain each). To build the MSI-2-improved database, T. interdigitale and T. indotineae strains were randomly selected from all the strains that were available for this study. A list of the references of the T. mentagrophytes complex of species can be found in Supplemental Table 1.”

Authors should carefully check the numbering of the references in the text. It appears that reference [18] should be reference [22] (Jabet et al.) on line 295.

Thank you to the reviewer for his/her careful reading of our manuscript that allowed to point out that the reference software was deactivated during part of the redaction of the manuscript. We carefully checked the reference numbering throughout the manuscript and noticed several other errors in the numbering and we corrected them.

Reviewer 3 Report

MALDI TOF mass spectrometry is a well-established method for identification of fungal or bacterial species. The challenge for Trichophyton indotineae identification is the close relationship to the species of the T. mentagrophytes complex, which make it difficult to distinguish between all kinds of putative subtypes. The improved database allows the correct species identification with a rate of 96% and a sensitivity of 99.6% depending also on approved growth conditions.

Minor points

Line 202:  T. indotineae needs to be shown in italics as species name

Table 1 and 2 T. metagrophytes means T. mentagrophytes?

Line 276:  itraconazone means itraconazole?

Author Response

Reviewer 3

MALDI TOF mass spectrometry is a well-established method for identification of fungal or bacterial species. The challenge for Trichophyton indotineae identification is the close relationship to the species of the T. mentagrophytes complex, which make it difficult to distinguish between all kinds of putative subtypes. The improved database allows the correct species identification with a rate of 96% and a sensitivity of 99.6% depending also on approved growth conditions.

Minor points

Line 202:  T. indotineae needs to be shown in italics as species name

We are sorry for overlooking this in the manuscript. T. indotineae is now in italics in this part of the manuscript.

Table 1 and 2 T. metagrophytes means T. mentagrophytes?

We thank the reviewer for his/her careful reading of the manuscript and the tables that highlighted this typo. The correction has been made in both tables.

Line 276:  itraconazone means itraconazole?

The correction has been made.

Reviewer 4 Report

Interesting study, well-designed. There are small details to be corrected, mainly with the names of the microorganisms and the acronyms. 

I suggest making a graph of the methodological design, in order to make the number of isolates, species and spectra used more understandable. 

Line 66. consider, To date, until now instead of Today.

Line 94 The acronym should be MALDI-TOF MS throughout the document, however there are sections where MALDI Tof is found e.g. line 99.

Line 99: meaning of the acronym MSI

Line 107: assign the acronym to Sabouraud Agar example SDA and describe the meaning of the acronym IDFP.  In the discussion part the acronym for Sabouraud GC is used, please do it from the methodology.

Line 118: consider the word stage instead of age (many parts in the document).

Line 127. T. indotineae. The genus name should only be written on the first mention. Check this throughout the document.

line 202. italics 

line 204. remove italics from reference.

Table 2: please add the value in (%) of the analyses.

Figure 2. Adjust the scales of the two graphs. Are you sure it is percentages that represent the y-axis?

Table 3 delete the word peak, 

Overuse of the word use in the conclusion.

Round 2

Reviewer 1 Report

The second version of the manuscript has significant improvements over the first one. The comments and requests were carefully addressed by the Authors. The manuscript is very close to the final form, but please fix the first column of Table 3, where upright and slant text is mixed.

When preparing the final version, the Authors can consider embedding figures as vector graphics in the MDPI template. For this purpose, WMF file format worked in my hands, though it is old and has issues. Otherwise there is a chance of getting insufficient quality of published PDF file.

Author Response

The second version of the manuscript has significant improvements over the first one. The comments and requests were carefully addressed by the Authors. The manuscript is very close to the final form, but please fix the first column of Table 3, where upright and slant text is mixed.

Thank you for noticing this error. We modified the 2 numbers that were in italics, and wrote them in slant text. 

When preparing the final version, the Authors can consider embedding figures as vector graphics in the MDPI template. For this purpose, WMF file format worked in my hands, though it is old and has issues. Otherwise there is a chance of getting insufficient quality of published PDF file.

Thank you for the information. We checked that our figures are 600ppi, which should be enough for publication, but if not, we will try to transform them as vector graphics. 

Reviewer 4 Report

I have no further comments, the document is ready for publication.

Author Response

I have no further comments, the document is ready for publication.

We thank the reviewer for his/her kind comment and his/her thorough reading of the manuscript.